# A Comprehensive Approach in Medical Nutrition Therapy for Adults' Weight Loss Management in Lebanon

**Marie-Therese Khalil [1,2], Joseph Matta [3,4], Mateja Videmšek [5], Damir Karpljuk [5] and Maja Meško [6,7,*]**

1 Ministry of Economy and Trade, Riyad el Soleh St, Azarieh Bldg., Beirut PO Box 11-1520, Lebanon; mt.k@live.com
2 Faculty of Organization Studies Novo mesto, Ulica talcev 3, 8000 Novo mesto, Slovenia
3 Industrial Research Institute, IRI Bldg., Lebanese University Campus, Hadath (Baabda), Beirut PO Box 11-2806, Lebanon; chem@iri.org.lb
4 Faculty of Pharmacy, Department of Nutrition Université Saint Joseph, Beirut PO Box 11-5076, Lebanon
5 Faculty of Sport, University of Ljubljana, Gortanova 22, 1000 Ljubljana, Slovenia; mateja.videmsek@fsp.uni-lj.si (M.V.); damir.karpljuk@fsp.uni-lj.si (D.K.)
6 Faculty of management, University of Primorska, 6000 Koper, Slovenia
7 Faculty of organizational sciences, University of Maribor, Kidričeva 55a, 4000 Kranj, Slovenia
* Correspondence: maja.mesko@fm-kp.si

**Abstract:** The objective of the research is to identify the different factors of Lebanese culture that interfere with weight loss therapy and assist the field of nutrition in homogenising in a standardised manner the protocol of Medical Nutrition Therapy (MNT). The first part of the study is based on a literature review, and, in the second part, quantitative analysis was used. The research was conducted on 514 Lebanese adults via questionnaire. The analysis was performed with the AMOS (Version 22, IBM®, Amonk, NY, USA) statistical tool. For the analysis of correlations, chi-square and non-parametric tests were used. Variables affecting weight loss management were identified with the aid of seven hypotheses using structural equation modelling (SEM). Body shape and Body Mass Index (BMI) were found to be inter-related to cognitive behaviours toward food, lifestyle practices, medical conditions, food and beverages. In parallel, and based on the research results, younger adults, in particular women, have better BMI and look better in terms of body shape. Ageing has a direct impact on weight gain. Older people have a lower activity level, which is more prevalent among women, and they also prefer to eat typical Lebanese food. Habits, such as smoking, drinking alcohol, are directly related to obesity and some medical conditions. Low physical activity influences the problems related to body shape. For further studies, one should also include types of physical activities in terms of intensity and number of hours. This would assist the study in being more specified and credible toward the effect of exercise on weight loss management.

**Keywords:** medical nutrition therapy; weight loss management; cost-effectiveness; anthropometry; culture; stress; physical activity

## 1. Introduction

Theoretical literature is crucial to building a strong assumption of the model structure by gathering facts from previous research studies and reviews about each mechanism related to weight management. Alternatively, elements and variables that actively interfere with Medical Nutrition Therapy (MNT) are to be determined based on their effect on obesity financially and economically. The gathered

assumptions in terms of variables are used to build a series of hypotheses for which a questionnaire addressed to a pool of Lebanese adults shall be compared statistically to check the validity of each hypothesis and its application in Lebanon.

Worldwide obesity has more than doubled since 1980. World Health Organization (WHO) figures indicate that obesity is spreading around the world as a "global epidemic". However, the proportions vary from country to country and within geographical areas in the same country [1,2]. Nowadays, excess weight or obesity affects 30–70% of the general population. In recent years, Lebanon has been experiencing a nutritional transition in food choices from the typical Mediterranean diet to the fast-food pattern.

As a consequence, the dietary habits of young adults have been affected; thus, overweight and obesity are increasingly being observed among the young [3]. In Lebanon, similar to other Eastern Mediterranean countries, the rate of overweight and obesity among adults is reported to follow an escalating secular trend, with its prevalence estimates increasing from 54.4% to 65% over the past decade [4]. In a more recent study, Lebanese body mass index (BMI) above the normal range was estimated to be 62.8% (men 67% and women 58.7%) among which 28.2% are obese, which is alarmingly high [5].

The growth of overweight problems could be justified by either a higher intake of calories or a lower expenditure of calories in daily activities [6], assuming that obesity is the result of a positive energy surplus between calories consumed and calories expended, resulting in excess body weight. Exposure to palatable food temporarily also inhibits thoughts about dieting in chronic dieters and leads to overeating [7]. Alternatively, obesity was found to be associated with an increased risk of all-cause and cardio-vascular disease mortality, raising the risk of morbidity from hypertension, dyslipidemia, type 2 diabetes mellitus (diabetes), coronary heart disease, stroke, gallbladder disease, osteoarthritis, sleep apnea and respiratory problems, and some cancers [8].

According to prior literature, between 15% to 20% of people in the general population are only successful at long-term weight loss maintenance, which is defined as being intentionally losing and keeping the loss of ≥ 10% from initial weight for ≥ 1 year [9–12]. Reasons attributed to diet failure could be linked to hormones and other homoeostatic, environmental and behavioural factors, but the most common reason is attributed to stress and food deprivation [13–16].

This low percentage of successful weight loss is also applied in Lebanon [17]. It could be improved when setting a customised protocol in Medical Nutrition Therapy (MNT), which addresses individual nutrition needs based on personal and cultural preferences, health literacy, and access to healthful foods. The goals of MNT are to promote and support healthful eating patterns, emphasising a variety of nutrient-dense foods in appropriate portion sizes to achieve/maintain body weight and health goals. MNT is a valuable adjunct to health management programs that can be implemented for a relatively low cost [18]. By using a medical model, clinicians can provide more proactive and effective treatments in assisting their patients with weight loss [19].

Noting that anthropometric indicators used during general assessment of overweight and obesity are based on Body Mass Index (BMI), neck and waist circumferences and body shape type, which favours the apple shape for obese individuals.

In general, some interventions in daily behaviours and lifestyle may affect weight loss in a positive or negative pattern such as stress, physical activity, detox, professional assistance, image satisfaction, smoking, alcohol consumption, sleeping, meal frequency, home cooking, non-invasive slimming machines, food intolerance, supplements and medications for weight loss.

Stress, which is defined as 'the generalised, non-specific response of the body to any factor that overwhelms, or threatens to overwhelm, the body's compensatory abilities to maintain homeostasis' [20], can be either acute (suppress appetite) or chronic (stimulate appetite) [21,22]. As for physical activity, a combination of increased energy expenditure through exercise and reduced energy intake is likely to produce more significant weight loss and more favourable changes in body composition than either exercise or diet alone [23,24]. Exercise is widely regarded as one of the most valuable components of



behaviour that can influence body weight and therefore help in the prevention and management of obesity [25].

Obesity is a global public health problem and a risk factor for several diseases that financially impact healthcare systems and impose a considerable financial burden on the state [26]. In this context, annual costs are being attributable to medical expenditures, presenteeism and absenteeism [27]. The costs associated with absenteeism are probably higher than most managers would expect [28]. Obesity imposes a considerable financial burden on states, accounting for 6.5% to 12.6% of total absenteeism costs in the workplace. As a result, obesity, but not overweight, was associated with a significant increase in workdays absent, from 1.1 to 1.7 extra days missed annually compared with normal-weight employees. Alternatively, potential risk factors contributing to presenteeism include being overweight, a poor diet, a lack of exercise, high stress, and poor relations with coworkers and management [29]. On the other hand, consuming a high-quality diet and engaging in moderate levels of physical activity was negatively associated with absenteeism and reduced expected frequency by 50% and 36%, respectively [30].

The empirical evidence has shown beyond doubt that obesity negatively impacts individuals, healthcare systems, employers and the economy as a whole [31]. Organisations must create a work environment that supports high-quality performance, as well as individual and organisational health, development and growth. Employee health needs include weight management, improving fitness and nutrition in terms of eating habits, and decreasing coronary risk.

The head of the nutrition department at the Ministry of public health in Lebanon as well as the head of the syndicate of dietitians in Lebanon was contacted to check for an update and recent figures in the domain of obesity and the situation of healthcare systems under treatment and also its effect on the work efficiency and performance of the employees. Unfortunately, no official data are available for the present time on the situation of healthcare systems in the domain of nutrition and no studies were even performed to monitor the effect of obesity on absenteeism or presenteeism in Lebanon and its cost on different levels.

The research is mainly addressed to Lebanese society. It aims to develop a model setting the guidelines for a scientific assessment and data analysis followed by professionals and practitioners in the dietetic field. By gathering and identifying the relative parameters and variables, one can build a protocol that directs the MNT in a way to increase compliance of the patients towards diet. This guideline aims at providing useful information on how to achieve weight reduction and maintenance of healthy body weight in particular for the Lebanese society. Besides, this study provides an informative database regarding the Lebanese wellness status and health issues in nutrition in terms of lifestyle conditions, mainly physical activity, traditional influences and sociocultural approaches. Therefore, Lebanese scholars and professionals in healthcare and health life domains could take advantage and benefit from the findings to direct their assessments in the fields related directly or indirectly to nutrition and dietetics.

## 2. Materials and Methods

### 2.1. Hypothesis

At first, the theoretical background for the research is built according to literature reviews on MNT concerning weight loss management from different perspectives. Consequently, data are gathered based on which, brainstorm meetings with professional dietitians and practitioners are made, and seven hypotheses came out as follows.

Hypothesis 1: Young people, in particular women, tend to look better physically in terms of body shape than older people; as one gets older, the apple shape becomes more developed. Physical activity, which is less common among women, decreases consecutively with ageing.

Hypothesis 2: Women at any age have higher levels of body image dissatisfaction than men.

Hypothesis 3: Active people have a better shape, medical conditions, and thus they lose weight faster.

Hypothesis 4: Emotional eating interferes negatively with Medical Nutrition Therapy. Stressed people do not lose weight easily. Stress is directly related to appetite increase and fat deposit.

Hypothesis 5: People suffering from medical problems and smoking/alcohol consumption have a greater tendency to be above average weight. Medical problems, especially those related to Gastrointestinal problems might implicate possible food intolerance/allergy.

Hypothesis 6: Users of dietary supplements are more likely to be educated and have better health and wellness patterns than non-users.

Hypothesis 7: Older people, especially women, have better attitudes towards healthy food, healthy eating habits, intention to change diet.

### 2.2. Questionnaire and Data Collection

The questionnaire has 125 variables and was composed of demographic information, dietary and anthropometric information, lifestyle, medical information, and family history. The survey is conducted based on the questionnaire; its validity is checked using a pilot study (n = 50), which was run in November 2016, while its reliability is checked using the Cronbach alpha coefficient, which is computed and found to be 0.720; reliability is considered achieved when the coefficient exceeds the recommended value $\alpha = 0.7$, which means that the questionnaire is consistent and reliable.

The demographic collected data in terms of percentage are compared between the pilot study and the entire survey (see Table 1). As a result, all the variables are nearly the same except for the age ranges, which are higher for the youngest population compared to the entire population. The address of the participants is divided between city, village, and mountain, differing on the geographical scale. The city is on the coast, and the mountain is where it snows in winter, and the village is in between.

Feedback and observed issues were considered, and corrections were added (Arabic language, representative pictures, some explanations and examples, font and layout). So, when re-piloting and based on prior feedback, participants are informed that the survey would take between 10 to 15 min to be filled. The questionnaire was updated and conducted by eight registered dietitians on a one-to-one interview basis during the 5-month period between February 2017 and June 2017. Each interviewer had between 60 and 70 copies of the questionnaire, and since data collection was conducted on site, there were no missing data. Interviewers were picked from the neighbourhood, workplace and even from clinics. Participation in the survey was anonymous, voluntary and followed the ethics as stated and written on the header of the survey, in which the estimated time for finishing it (10 to 15 min) was also written on the top of each questionnaire.

### 2.3. Tests Performed on Collected Data

The research model is carried out through structural equation modelling (SEM) and confirmatory factor analysis (CFA). Since the questionnaire had 125 variables, we applied the variables reduction method using principal component analysis (PCA) in SEM to define the number of components under which the variables could be grouped. Variable reduction and specifying component categories are made through CFA; the determinant was found to be acceptable with a value of 0.01, and KMO (Kaiser–Meyer–Olkin) was found to be acceptable with a value of 0.756, indicating a 'moderately good' sampling adequacy for the analysis according to Field [32]. It is worth noting that all KMO values for individual items were above the acceptable limit of 0.50 [32] except for the last component, which was close to 0.50. Bartlett's test of sphericity $\chi^2$ (276) = 3379.834, $p < 0.001$, indicated that correlations between items are sufficiently significant for PCA. An initial analysis was run to obtain eigenvalues for each component in the data. Thirty-seven components had eigenvalues over Kaiser's criterion of 1 and in combination explained 61.08% of the variance. Given the large sample size, and the convergence of the scree plot and Kaiser's criterion, seven components were extracted with a value of a factor greater than 0.40; this is the number of components that were retained in the final analysis. The model fit index

of CFA is calculated relative to the total sample, normal-BMI category and over-BMI category. All fit indexes were found within the normal range.

**Table 1.** Demographic data of the pilot study (N = 50) and the entire survey (N = 514).

|  | Pilot Study | Entire Study |
|---|---|---|
| **N** | 50 | 514 |
| **Gender** | | |
| Male | 50% | 43.6% |
| Female | 50% | 56.4% |
| **Religion** | | |
| Christian | 58% | 64% |
| Muslim | 40% | 31% |
| **Age** | | |
| 19–27 | 18% | 35.2% |
| 28–36 | 34% | 23.2% |
| 37–45 | 14% | 16.9% |
| 46–54 | 18% | 13.2% |
| > 55 | 16% | 11.5% |
| **Work** | | |
| No | 26.5% | 26% |
| Part-time | 12.6% | 10% |
| Full-time | 60.9% | 64% |
| **Monthly income** | | |
| ≤ 500 USD | 22% | 31.1% |
| 501–1000 USD | 32% | 23.2% |
| 1001–1500 USD | 20% | 20% |
| 1501–2000 USD | 12% | 12.1% |
| ≥ 2001 USD | 14% | 13.6% |
| **Education level** | | |
| Basic-elementary | 14% | 9.1% |
| High school | 1% | 19.6% |
| Technical level | 14% | 12.1% |
| Bachelor degree | 28% | 32.5% |
| Master's degree | 24% | 23% |
| Doctorate | 2% | 3.7% |
| **Home address/work address** | | |
| Mountain/mountain | 4% | 7.2% |
| Mountain/city | 16% | 15.4% |
| Village/village | 10% | 13.6% |
| Village/city | 16% | 17.7% |
| City/city | 54% | 46.1% |
| **Civil status** | | |
| Independent single | 16% | 12.1% |
| Single with parents | 32% | 37.2% |
| Concubine | 2% | 1% |
| Married | 42% | 45.9% |
| **Divorced/widowed** | 8% | 3.9% |
| **Number of children** | | |
| 0 | 54% | 55.1% |
| 1 | 10% | 8.9% |
| 2 | 8% | 19.1% |
| 3 | 20% | 10.3% |
| ≥ 4 | 8% | 6.6% |

Data analysis was carried out by exploring each hypothesis and checking its validity using different statistical tests. Concerning the hypotheses, a cross-sectional study analysis was carried out applying SEM. For this purpose, the extended statistical program of SPSS (Version 22) called AMOS (Version 22, IBM®, Amonk, NY, USA) was used to test path dependency, and correlations between demographic variables and different components were retrieved from the questionnaire such as medical conditions, food consumption, lifestyle, behaviours toward food, and many other inter-related variables.

The questionnaire based on the theoretical background and literature scanning was developed, for which a pilot survey testing its internal consistency was made. Consequently, data on the empirical study were gathered in order to reach a comprehensive research sample for SEM.

After data collection, statistical analysis based on SEM was provided. The research model of participation was tested based on model fit analyses and reliability of data. Consequently, hypotheses testing was demonstrated.

Three statistical procedures were used in our quantitative research, namely, descriptive and analytical statistics, CFA, and SEM. Analysis framework of the collected data from respondents was analysed using Microsoft Excel 2016 and Statistical Package for Social Sciences (IBM SPSS statistics 22); statistical significance was assumed when $p < 0.05$. The primary method used was Pearson's chi-square test, other methods and tests used include one-way ANOVA, Mann–Whitney U test, Kruskal–Wallis H test.

Effect size for chi-squared or Cramer's $\Phi$ (Phi) also named Cramer's V is a correlation coefficient used to represent the association between two categorical variables: $\Phi = \frac{\sqrt{x^2}}{N\,(K-1)}$ [33].

The data analysis procedures can be visualised as follows (see Figure 1).

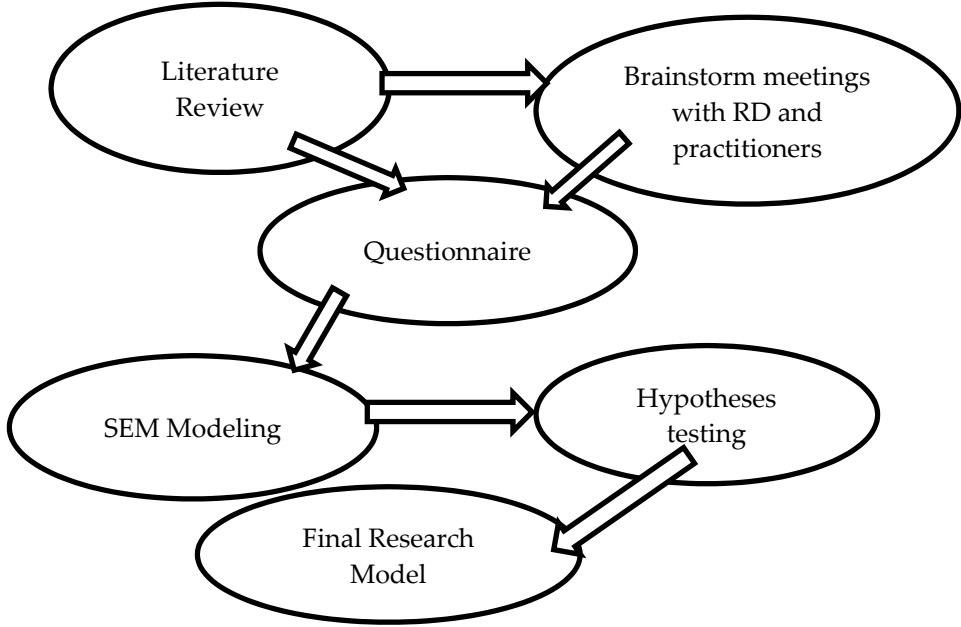

**Figure 1.** Model of data analysis procedures.

## 3. Results and Discussion

The sample was described, and the performed tests were defined. Consequently, data analysis procedures in the quantitative part are developed for which seven hypotheses are explained, discussed and verified if accepted or not. The percentage of dependent and independent variables was also defined, and variables affecting BMI were determined in terms of strong, moderate, weak (by mean of effect size) or no association at all.

The data collected from Lebanese adult people reflect a sedentary lifestyle. Even though BMI is normal for half of the population (50.8%), especially for women (12.6% males, 38.1% females), where an hourglass shape is mostly defined (76.6% of the hourglass shapes had a normal-BMI), the level of physical activity is considered low, where 62.8% of the participants do not engage in any indoor activities, 49.8% do not practice any outdoor activity, and 73.7% do not work outside/in the garden. These percentages increase with age as the apple body shape develops accordingly with a strong correlation: $y = +7.88x + 2.06$ ($R^2 = 0.9307$) (39% of the ones aged above 55 years have an apple-shaped body). A one-way ANOVA test was conducted to compare the means of the three forms of physical activity against age. Subsequently, there was a significant effect of age on indoor activity and gardening at $p < 0.001$ [$F_{(5,508)} = 4.264$, $p = 0.001$] and [$F_{(5,508)} = 4.093$, $p = 0.001$], respectively. However, there was no significant effect of age on outdoor physical activity [$F_{(5,508)} = 2.210$, $p = 0.052$]. In general, the proportion of active men is greater than women, especially for indoor and gardening activities, which are emphasised for respondents living in rural regions (82.3% of those who lived and worked in the city had no interest in gardening whereas around 56.8% of those living and working in the mountains showed interest). In this context, a non-parametric Mann–Whitney U test was conducted to determine whether there was a difference in physical activities between men and women. The results of that analysis indicated that indoor physical activity as well as gardening were statistically significantly greater for men than for women, [($z = -2.087$, $p = 0.037$) and ($z = -3.250$, $p = 0.001$)], respectively. In contrast, the difference in outdoor physical activity between men and women was not significant ($z = -0.207$, $p = 0.836$).

The apple shape, which is more prospected as an indicator of obesity, was directly related to BMI (92% of the apple shapes had an over-BMI), age, and was more revealed among men in general and women after menopause (25.4% of males have an apple shape compared to 10.3% of females). On the other hand, the most common stress effect detected among participants was related to financial issues (61.3%). Only men with over-BMI eat less when feeling angry or sad. The same effect was noticed among women who also feel anxious independently of their BMI.

The relation between BMI and the rest of the variables was studied, and the results showed that participants with over-BMI ate more food considered to provoke weight gain daily, such as yellow cheese (60%), butter (53.8%), regular mix nuts (60.7%). Nevertheless, they ate less raw nuts (72% of participants who never eat raw nuts have an over-BMI) and mayonnaise light (52.4% of participants who never eat mayonnaise light are over-BMI). As for the beverages, participants with over-BMI consumed more regular soft drinks (70% of those who consume more than 6 cups per week of regular soft drinks have an over-BMI) and alcohol in general (81% of those who drink more than 6 cups per week have an over-BMI). Another difference was detected in the portion size preference, where small portion size was more noticed among adults with normal-BMI who tend more to self-monitor their calorie intake along (80.6% and 100% of participants who prefer small and extra-small meals have normal-BMI whereas 72.7% and 75% of participants who preferred large and extra-large meals had over-BMI), with a fine selection of lower calories food ((around 60 to 70%). This same group of participants ate more slowly than over-BMI (65.7% of those who eat fast have an over-BMI and 59.3% of those who eat slowly have a normal-BMI) and ate less outdoor in frequency mode per week (69.8% of those who eat more than four times per week have an over-BMI). Besides, normal-BMI had less self-image satisfaction and exercised more indoors and outdoors. On the other hand, participants who were subject to psychotherapy were more prone to have over-BMI (Out of 5.4% of respondents who had a positive answer, 67.8% had an over-BMI). Usually, they slept less than those with normal-BMI (77.3% of those who sleep more than 9 h at night had a normal-BMI, whereas 71.4% of those who only sleep 3-4 h had an over-BMI). The over-BMI category was also found to be associated with smoking both cigarettes and a cigar (66.7% from those who smoke every day had an over-BMI).

In addition, over-BMI was found positively related to diabetes (79.2%), cholesterol/triglycerides (77.9%), blood pressure issues (63.6%), bloating (54.7%), edema or leg swelling (61.5%) and inflammation/infection problems (63.3%). Finally, a moderate association was found between BMI of

participants and their mothers' weight ($X^2(4) = 16.155$, $p = 0.003$, Cramer's $V = 0.177$) as well as with the frequency of family diseases; the percentage of participants with over-BMI against normal-BMI increases with the number of diseases, starting with a percentage of 39.8% to reach 80% for the participants who reported 6-7 diseases in their family. Irrelevant associations were found between BMI and religious lent commitment as well as with drinking regular water and coffee ($p > 0.05$).

The seven hypotheses that were analysed regarding the research subject aim at answering the main question regarding the variables directing Medical Nutrition Therapy (MNT) for adult weight loss in Lebanon, which are presented in the final illustration of each of the seven accepted hypotheses via AMOS software.

The seven hypotheses were defined and analysed via the chi-square test to check for any possible association, and the results are summarised in Table 2.

Since the thesis subject is about weight loss in Medical Nutrition Therapy, the main component that should be compared between variables is Body Mass Index (BMI) as an anthropometric indicator. For this purpose, the chapter "results" included, in addition to the descriptive statistics, the variables that mainly affect BMI through chi-square test analysis.

The discussion of the seven hypotheses noticed that women presented a lower BMI, which was positively related to age. A significant association was detected between BMI and body shape; in fact, 92% of the ones with apple shape had an over-BMI. Inversely, the ideal body shape for women (hourglass) was more relevant for the participants with normal BMI (76.6%) and the inverted triangle shape as well as the rectangular shape for men, both decrease in percentage when age increases. The shapes that are not favourable esthetically and on a health basis (apple and pear shapes) were less seen among active participants. Physical activity that was low in general was more favourable among men for the indoor and gardening types. On the other hand, outdoor physical activities had no preference among gender (Hypothesis 1).

Body-image satisfaction, which was relatively high among participants, was found inversely associated with BMI. Women, in particular, were more interested in losing weight than men. Ageing was directly related to body-image satisfaction. Respondents with high physical activity had a high body-image satisfaction and had rectangular and inverted triangle shapes for men and an hourglass shape for women. Other variables were tested for the body-image satisfaction/interest in losing weight between male and female, such as religion, number of children, marital and work status (no major significant association was detected except for part-time participants, who had a high practical significance for the body-image satisfaction, and the full-time workers showing a medium practical significance for the level of interest in losing weight, and the ones having two children but this time with a high practical significance) (Hypothesis 2).

Hypothesis 3, concerning physical activities and medical conditions, showed that apple and pear shapes were more detected among sedentary participants, while hourglass shape was more relevant for active women and rectangular as well as inverted triangle shapes for active men.

In general, active participants had less medical issues; health problems that were mostly manifested by respondents (more than (>) 30%) were diabetes, cholesterol/triglycerides, low/high blood pressure, bloating, low back pain, oedema, fatigue, anxiety, migraine/headache, cancer and cellulite.

On the other hand, the most relevant medical conditions that showed a significant inverse association with physical activities among respondents were:

Diabetes, cholesterol/Triglycerides, oedema, cancer inversely associated with outdoor physical activity.

Cellulite, anxiety, bloating associated with gardening activities.

Low/high blood pressure inversely associated with indoor physical activity.

Fatigue inversely associated with both outdoor and gardening activities.

Low back pain inversely associated with both indoor and outdoor physical activity.

Migraine/headache inversely associated with all type of physical activity.

**Table 2.** Summary of the seven hypotheses with their results in terms of acceptance.

| Hypothesis | Level of Acceptance | Remarks |
|---|---|---|
| Hypothesis 1: *Young people, in particular women, tend to look better physically in terms of body shape than older people; as getting older, apple shape is more developed. Physical activity which is less common among women decreases consecutively with ageing.* | Accepted for the first part. But, a significant effect of age was only observed on indoor activity and gardening. | - At an advanced age, both genders showed a tendency for a common apple shape.<br>- Females showed a better physical appearance and body shape, especially when young, although their level of activity was low in general and much lower than males. |
| Hypothesis 2: *Women at any age have higher levels of body image dissatisfaction than men.* | Rejected. | - Most of the respondents from both genders showed a tendency to favour self-body satisfaction with no significant association.<br>- No significant association was detected between self-image satisfaction and age groups and gender as well ($p > 0.05$). |
| Hypothesis 3: *Active people have a better shape, medical conditions and thus they lose weight faster.* | Rejected only for the outdoor sports activity. Gardening and indoor physical activity level were inversely related to ageing. | When increasing physical activity, one should stress on signs inversely related to it, such as fatigue, low back pain, mood swings, anxiety, bloating, Migraine/headache, constipation and digestive problems. |
| Hypothesis 4: *Emotional eating interferes negatively with Medical Nutrition Therapy; Stressed people do not lose weight easily. Stress is directly related to appetite increase and fat deposit.* | Partly accepted. Daily stress effect was linked to abdominal fat deposit and apple shape figure. | - Behaviour toward eating (self-monitoring of food and selected lower calories) associated with weight loss.<br>- Feeling angry, sad or anxious had a statistically significant linear relationship with BMI. Instead, happiness did not show any statistical correlation with BMI. |
| Hypothesis 5: *People suffering from medical problems and smoking/alcohol consumption have a greater tendency to be above the normal weight. Medical problems, especially those related to Gastrointestinal problems might implicate possible food intolerance/allergy.* | Partly accepted. smoking and alcohol consumption was found associated with BMI. | - A strong significance was found between alcohol consumption and heart conditions, diabetes, cholesterol/triglycerides and blood pressure.<br>- A small significant association was detected between indigestion/bloating and food intolerance. |
| Hypothesis 6: *Users of dietary supplements are more likely to be educated and have better health and wellness patterns than non-users.* | Rejected. No association was found between users of dietary supplements and their level of education and income. | A small association was found between supplements intake for general health being and the following medical conditions: heart conditions, diabetes, digestive problems, constipation, bloating, low back pain, oedema, anxiety, depression, mood swings, frequent colds or flu, inflammation/infection, and migraine/headache. |
| Hypothesis 7: *Older people, especially women, have better attitudes towards healthy food, healthy eating habits, intention to change diet.* | Rejected with a small exception. | Only old Lebanese favoured the "regular Lebanese menu" without even caring about their body-image. |

Emotional eating was not found significantly associated with BMI. The level of appetite when facing emotions was relatively constant or neutral for males except for those with over-BMI who tend to eat less when angry and/or sad. On the other hand, women tend to eat less when feeling angry, sad and/or anxious but without change of their appetite level when feeling happy like men did. Stress was more related to financial, work, and family issues. The apple body shape was found to be irrelevant with stress; emotional eating seems to control appetite. Most of the women who have stress eat less, and most of the men do not change their appetite. Therefore, the tendency to eat less and as a result have a decrease in BMI is more relevant when facing stress (Hypothesis 4).

BMI was found to besignificantly associated only with the following medical conditions; diabetes, blood pressure, bloating, oedema, inflammation/infection, cholesterol/Triglycerides. Participants who neither smoke cigarettes nor cigars had an average BMI; in contrast, participants who smoke cigarettes more often and drink alcohol had an over-BMI. An association with medium significance was only found between smoking cigarettes and the following medical issues; heart conditions, diabetes, cholesterol and blood pressure. Alternatively, bloating, as well as indigestion, were found to be directly associated with food sensitivities but with a small significance effect (Hypothesis 5).

Supplement intake was linked neither to the social status nor to BMI. However, when computing against medical conditions, some relevant associations were found inversely related with small significance between supplements intake for general health being and heart disease, diabetes, digestive problems, constipation, oedema, depression, frequent colds/flu, inflammation/infection, migraine/headache. A significant association was detected between supplements for weight loss and constipation, oedema, depression, mood swings, hormone issues, migraine/headache and cellulite (Hypothesis 6).

No relation was found between older participants and portion size preference, eating pattern, and cognitive behaviours toward food. Instead, older people tend to prefer a regular Lebanese menu or simply present no preference at all. Older participants were less likely to plan their meals in advance, count their calories and select them accordingly. Besides, old participants showed low self-image satisfaction compared with younger ages. It is worth noting that females present positive cognitive behaviour toward food, and prefer light food and tend to eat slowly to moderate compared with males. Moreover, females were more likely to be healthy in terms of food and beverage choice. On the other hand, aged respondents were also found to be healthy in choosing food items and also when drinking (Hypothesis 7).

Considering theoretical literature, all reviews and scientific studies from WHO agreed on the fact that obesity percentage is increasing worldwide at an enormous scale, presenting an issue that affects everyone and harms general health and well-being. Theoretically, elements that present a negative impact on weight loss, contributing to a possible failure in weight management, are chronic stress, low physical activity, alcohol consumption in a heavy way, food intolerance, late bedtimes, and short/long hours of sleep. Other elements, such as acute stress, smoking, cooking at home, meal replacements, some supplement intake and Chinese herbs related to detoxification, a balanced Mediterranean diet in addition to a consistent role of assistance followed by dietitians on setting a defined regimen of meal frequency, and a constructed physical activity system, should all have a positive effect on weight loss and its maintenance. Some points listed in the theoretical part do not affect obesity and weight loss, like, for instance, consuming alcohol in moderation, non-invasive slimming machines, supplements and drugs intended for weight loss use.

In our study, the anthropometric indicators that were only considered were BMI and body shape type. Waist and neck circumferences were not measured. The over-estimation of BMI due to a possible increased lean mass weight can be neglected since, in our study, the majority of the respondents followed a sedentary lifestyle.

The apple shape, which is found predominantly for women at an advanced age (after 55 years old), could be linked to the hormonal imbalance that usually follows menopause, as many research had already proven.

Although, theoretically, women show more interest in losing weight than men do according to Grogan, our findings did not indicate a significant difference. There is a negative correlation between body-image satisfaction and BMI as well as with ageing between gender.

It is worth noting that a highly significant association was detected between gender for the body-image satisfaction of the participants working part-time. Contrastingly, medium to high significance between gender was found for the ones interested in losing the weight of the married participants having two children and working full time. Theoretically and according to many studies, such as the ones led by Aschbacher, stress is related to abdominal fat deposit leading to an apple body shape. In contrast, in our research, stress was not found to be associated with apple body shape even for stress emerging from studying, an opposite effect was observed. Participants with stress had less an apple shape than those with no stress. Emotional eating and behaviour toward food have a more substantial effect on manipulating BMI than stress does by themselves. Participants who were asked about the effect of stress said that they ate less as a consequence; this is why their BMI was found to be lower than expected.

Some bad habits, such as smoking cigarettes and consuming alcohol, were found to be linked to obesity in the research findings. Though smoking was theoretically linked to appetite decrease, its cessation lead to possible obesity according to many articles and authors, such as Chen and al. and Rupprecht.

On the other hand, the effect of hours of sleep linked to weight management was found similar to the literature review. Likewise, supplements intake was found, like in the theory, to be unlinked to weight loss, unlike supplements intended for detoxification and general well-being, which showed a significant effect on weight loss management.

Nevertheless, many theoretical topics were neither validated nor verified in the study, such as the role of dietitians in assisting the patients during their weight loss, detox techniques, the effect of meal replacements in therapeutics, in addition to the effect of non-invasive slimming machines, and some anthropometric measures like waist and neck circumferences.

The hypothesis was at first represented with a model illustration, then analysed to check if it was accepted or not.

Hypothesis 1 was accepted and verified.

Hypothesis 2 was tested but, at first, rejected unless another variable that has a similar meaning was engaged to check the consistency of the results. Therefore, when using the other variable, the hypothesis was accepted. Other variables were also checked to test if they have to be considered.

Hypothesis 3 was also accepted, but medical conditions associated with physical activity are categorised and retrieved.

Hypothesis 4 was partly accepted mostly for emotional eating, but the effect of stress on fat deposit was not validated. Therefore, the hypothesis accepted can be stated as follows: "*Emotional eating should be assessed with MNT and was found directly related to appetite change*".

Hypothesis 5 was found accepted only for alcohol consumption and smoking cigarettes and for some medical conditions.

The first part of Hypothesis 6 concerning education was rejected, and the second part concerning the effect of supplements on medical conditions was partly accepted.

Hypothesis 7 was partly rejected (concerning the age). Therefore, the accepted hypothesis could be said as follows: "Older people especially women, eat healthier food but only women have better attitudes towards healthy eating, healthy eating habits, intention to change diet".

The final research model summarizing the accepted results of the hypothesis is represented in Figure 2, in which the set of variables were grouped into five categories.

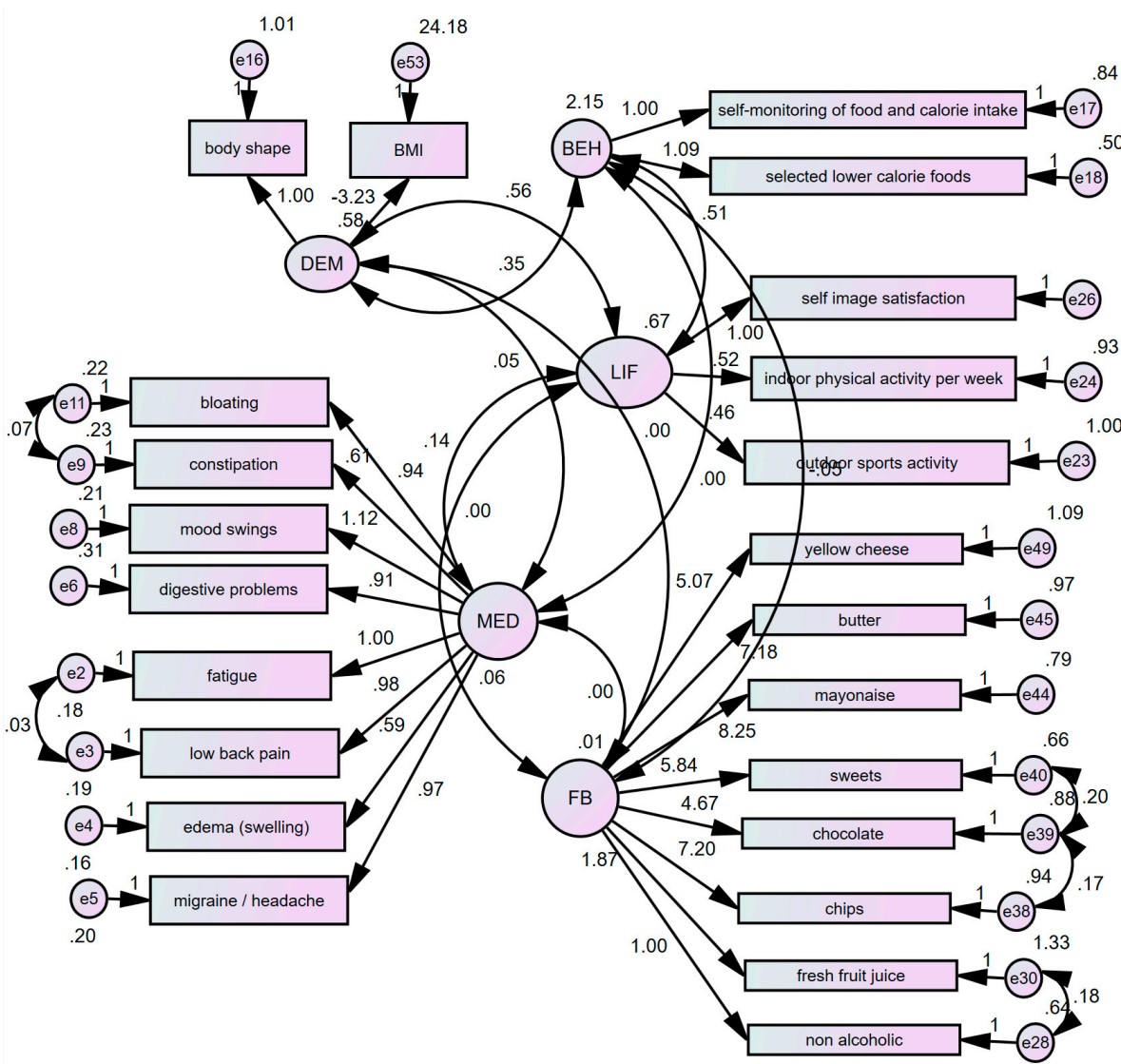

**Figure 2.** Modified Structural Equation Model. (N = 514. chi-square = 399.330; degrees of freedom df = 215; *p* < 0.001; e = error).

The goodness of fit of the model was tested, and the first set of statistics suggested that the fit of the data to the hypothesised model was not entirely adequate. Therefore, the model, through incremental fit indices, was considered to be unacceptable.

Based on the results of model estimation and fit, model modification analysis was run to adjust the model by adding covariates between the highest modification indices (MI). After adding the covariance, the standardised residual covariance matrix was added for signs of relationships that were not well explained by the model. Method of elimination based on Standardised Residual Covariance was used. For this purpose, standardised residual covariance less than 2.0 was eliminated. Therefore, a latent exogenous variable which appeared to be statistically insignificant, and its indicators had significantly high standardised residual covariance coefficients removed. The model modified is shown in Figure 1. Respectively, SEM with the following results on regression weight coefficients showed that all indicators were statistically significant (*p* < 0.05).

The modified model, with 276 distinct sample moments, 61 distinct parameters to be estimated, and 215 degrees of freedom. Improvement in chi-square coefficients (1.857), incremental, or, more importantly, in RMSEA analysis of absolute fit were observed. Results on comparative fit indices showed more plausible coefficients of 0.822 for NFI, 0.891 for TLI, and 0.907 for CFI. Respectively, we

reached closer to the adequate threshold of 0.90 for comparative fit indices. As for the Root Measure of Standard Error Approximation (RMSEA), the new modified model has improved to a desirable coefficient of 0.041 (lower than 0.05).

All reviews and scientific studies and analysis from World Health Organizations and led by different authors, including Kushner, Swinburne, Wolfe, Fujioka, Jensen, and Afshin, agreed on the fact that the obesity percentage is increasing worldwide on an enormous scale, presenting an issue that affects everyone and harms general health and well-being.

When analysing the findings and the figures, some elements related to physical activity, stress, smoking and alcohol consumption, sleep quality, supplement intake as well as other factors were found to be inter-related:

- Chronic stress;
- Low physical activity;
- Heavy alcohol consumption;
- Food intolerance;
- Late bedtimes;
- Short and long hours of sleep.

These stated components were found to have a negative impact on weight loss, contributing to a possible failure in weight management.

However, acute stress, smoking, cooking at home, meal replacements, some supplement intake and Chinese herbs related to detoxification, a balanced Mediterranean diet in addition to a consistent role of assistance followed by dietitians on setting a defined regimen of meal frequency, and a constructed physical activity system, should have all theoretically a possible positive effect on weight loss and its maintenance.

Some points that were listed in the theoretical part were shown to have no effect on obesity or on weight loss, such as consuming alcohol in moderation, non-invasive slimming machines, as well as supplements and drugs intended for weight loss use.

As for the anthropometric indicators, when assessing a patient in general, many studies proved that the most important indicator of overweight and obesity is his waist circumference followed by the ratio with his hips circumference which shows the body shape type. Furthermore, it was clear that there is a correlation between obesity and neck circumference, according to many authors and scientists, including Aswathappa, Clombo-souza and Hingorjo. However, since there is no official and unified measure, this parameter could be considered when following up the patients to determine their progress in losing weight next to their weight and waist circumference when lacking the right device for body composition which shows the fat content and its progress with time. In our study, we only considered BMI and body shape type and did not measure the waist and neck circumferences.

BMI that indicates the weight status relative to the height can be overestimated when applied to an athletic patient with a highly developed lean mass. Otherwise, it helps to determine the type of body, ranging from underweight to obese going through normal and overweight levels. However, because, in our study, the majority of the respondents followed a sedentary lifestyle, any potential over-estimation of BMI due to increased lean mass weight can be neglected.

Our data analysis showed that body shapes for people who have no physical activities are highly concentrated for apple and pear shapes, which are believed theoretically to be a fat deposit indicator for men and women. At the same time, however, that does not mean that people who practice more have better body shapes. The reason is that the sample for whom the analysis was performed was relatively lowly active, so the results for the opposite hypothesis are not significant nor reliable.

The apple shape, which is more predominant for women at an advanced age ($\geq$ 55 years old), can be linked to the hormonal imbalance that usually follows menopause, as many studies had already proven.

Although women show more interest in losing weight than men do, according to Grogan, our findings did not indicate a significant difference between both genders when it comes to the body-image satisfaction. The youngest women were found to be more satisfied with their body shape, especially those having a normal BMI. There is a negative correlation between body-image satisfaction and BMI as well as with ageing between gender, except for men showing a low level of body-image satisfaction; in this case, a weak correlation was found with ageing.

It is noteworthy that a significant high association was detected between gender for the body-image satisfaction of the participants working part-time. Medium to high significance between gender was found for those interested in losing weight among the married participants having two children and working full time.

Theoretically and according to many studies such as those led by Aschbacher, stress is related to abdominal fat deposit, leading to an apple body shape, whereas, in our research, stress was not found to be associated with an apple body shape except for stress emerging from studying, but this time with an opposite effect; in fact, participants with stress are less likely to have an apple shape than those with no stress.

Emotional eating and behaviour toward food have a stronger effect on manipulating BMI than stress does by itself. For instance, theoretically, chronic stress leads to obesity, but emotional stress seems to be more important to study and examine. Participants who were asked about the effect of stress said that they ate less as a consequence; this is why their BMI was lower than expected.

Some bad habits, such as smoking cigarettes and consuming alcohol, were found to be linked to obesity in our research findings. Smoking was theoretically linked to appetite decrease, and its cessation leads to possible obesity according to many articles and authors, such as Chen et al. and Rupprecht.

In contrast, hours of sleep were theoretically linked to weight management. Similarly, supplement intake for weight loss was also, as in theory, not linked to weight loss, showing no effect of its indication of use, unlike supplements for detox and general well-being, which showed a significant effect on weight loss.

Nevertheless, many theoretical topics were neither validated nor verified in our study because they were beyond the quantitative analysis and needed further implications. The subjects concerned the role of dietitians in assisting the patients during their weight loss, detox techniques, the therapeutic effect of meal replacements, in addition to the effect of non-invasive slimming machines, and some anthropometric measures, such as waist and neck circumferences. These need specific research aiming at a population following a strict diet, and the research should be comparative, and the data should be taken during the therapy; afterwards, its effectiveness to should be checked, even after more than two years, to monitor the success whenever present.

## 4. Conclusions

The subject of the research was set from the research question that consisted of finding the variables that interfere and have an effect on whether positively or negatively on weight loss management therapy.

The MNT protocol for adult weight loss management in Lebanon is new in the nutrition field through its original approach in making the assessment simple and easy to follow as a standard. The most crucial purpose of this research is to raise and improve the percentage of success in weight loss and to lessen the epidemic status of obesity. Besides, the aim of this research serves the dietetic field to become more effective, efficient and reliable for patients, dietitians and other related practitioners and MD.

Healthcare units in Lebanon can improve their patients' outcomes in terms of reducing obesity rate by providing dietitians with sufficient time, knowledge, empowerment and training based on the appropriate weight loss management model and nutrition guidelines. The emerging modern lifestyle, with its stress on time, changing eating habits, lack of physical activity and busy families with long working hours, puts a lot of pressure on the food system to meet dietary needs. Furthermore,

eating habits were associated with leading causes of illness and death, such as cardiovascular disease, some cancers, diabetes, hypertension and obesity, a fact that raises the stakes for the dietetics profession.

This study is mainly useful for the Lebanese community through highlighting the therapeutic conditions useful not only for the dietetic field in Lebanon to become more efficient but also for other similar cultures to benchmark as well.

The research showed a pattern that reflects an unhealthy lifestyle in the context of physical activity, which is one of the factors that may contribute to the fat deposition and distribution in the body. In order to maintain good well-being and prevent problems in health, one should pay attention to his body shape and increase his physical activity, especially when getting older.

The contribution of this study emphasises the need for the community to maintain and enhance a healthy lifestyle by focusing more on their level of physical activities and consider the ecosystem by caring more for the environment and preserving it.

The limitations for our research concerned the type of diet, which was mainly the Mediterranean. The time frame when the data were collected and analysed was between 2017 and 2018. Later on, the analysis and the written results, as well as the rest of the research, took place during 2019. Moreover, the research was conducted in Lebanon, particularly in its capital Beirut and another adjacent region. The participants were gathered in these areas although they were picked up, only Lebanese citizens were considered. Therefore, the findings reflect a common culture and a unified society. Only adults aged above 19 years old were considered in this study. The essential two religions considered were Christianity (64%), and Islam (31.1%). Even though the percentage was not as the real theoretical one (Muslims form up to 54% of the population and Christians 41%), most of the answers were gathered from Christians, limiting the effect of other religions in the findings. In this context, more Muslims should be included in the sample studied.

Other research parts not covered were:

- All types of diet other than the Mediterranean one.
- Lab testing Procedure for examining the hypothesis.
- Some categories were out of scope, such as, for instance, pregnant and lactating women. The sample also did not cover children and adolescents, and people who want to gain weight were also not included in the research since the guideline concerns only assistance in losing weight.

Further studies aiming at considering absenteeism in healthcare and other organisations to check the effect of obesity and its cost should be measured.

Further research should consider the assessment of the effectiveness of an MNT protocol for adults' weight loss in Lebanon and describe the nutrition assessment and intervention activities of dietitians.

**Author Contributions:** Supervision, J.M.; Writing–original draft, M.-T.K.; Writing–review & editing, M.V., D.K. and M.M. All authors have read and agreed to the published version of the manuscript.

**Funding:** This research received no external funding.

**Conflicts of Interest:** The authors declare no conflict of interest.

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
