# Peer review of "A Comprehensive Approach in Medical Nutrition Therapy for Adults’ Weight Loss Management in Lebanon"

_applsci, doi:10.3390/app10186600_

Round 1

Reviewer 1 Report

Review of the Manuscript applsci-922188

Title: A model of a Comprehensive Approach in Medical Nutrition Therapy for Adults’ Weight Loss Management in Lebanon

Dear Authors,

The study focused on the problem of the management of excessive body mass among Lebanese adults. The paper is original and I appreciate the work in conducting the research (with an international team) and writing the article. However, the structure of the article is closer to a students’ thesis than to a scientific paper. In my opinion, it cannot be published in this form. It should be rewritten in accordance with the guidelines for the author and submitted again.  

Please consider the following comments in rewriting the manuscript:

The title:

  • The title of the article does not correspond to its content. It suggests to provide a comprehensive approach for the management of excessive body mass among Lebanese adults, meanwhile it provides very modest information on.

The study goal was established as follows:

(lines 22-24) The objective of the research is to identify the different factors of the Lebanese culture that interfere with weight loss therapy and assist the field of nutrition to homogenize in a standardized  manner the protocol of Medical Nutrition Therapy (MNT);

(lines 64-67) The research is mainly addressed to Lebanese society and aims in developing a model setting the guidelines for a scientific assessment and data analysis followed by professionals and practitioners in the dietetic field.

(lines 528-529) The subject of the research was set from the research question that consisted of finding the variables that interfere and have an effect on whether positively or negatively on weight loss management therapy.

  • please formulate clearly the purpose of the research.

Featured Application:

  • even though Authors are encouraged to provide a concise description of the specific application or a potential application of the work, the corresponding fragment (lines 15-21) is not necessary and does not add anything to the article.

Keywords:

  • please consider whether words standard, protocol, guidelines and healthcare are related to the topic of the article.

Introduction and Literature review

  • these two sections should be joined, shortened and rewritten. Please add some information on the prevalence of overweight and obesity among Lebanese population.

Materials and Methods

  • this section should be divided into subsections, including e.g.: Study Design, Participants, Questionnaire, Statistical Analysis;
  • a flow chart of the study would be very helpful in understanding the course of the experiment;
  • methods are not described with sufficient detail, especially the questionnaire. For what purpose was the questionnaire used? Please provide more information about the structure of the study questionnaire and individual items, as well as the method used for its validation. Please confirm the time of filling the questionnaire (10-15 minutes) containing 125 variables (lines 135 and 143).   
  • there is no information about the study population. The characteristics of the study population should be added in table form.
  • all abbreviations should be explained when they appear for the first time in the text, e.g.: SEM (line 160) – explanation line 175;

Results

  • this section does not contain the results, but rather a discussion of the non-presented results. Please provide the research results based on the data obtained from the questionnaire.
  • what was the purpose of formulating the research hypothesis and how does it relate to the development of the MNT model?

Discussion

  • Discussion and conclusion do not refer to own results and the title of the study.

In conclusion, the article should not be published in this form. It requires many amendments regarding the purpose as well as the methodology, discussions, results and conclusions

Author Response

Dear Reviewer,

thank you for your comments. We recognise the value that is added.

We changed the title as you suggested. 

We formulate clearly the purpose of the research.

Keywords are checked. 

Introduction and literature review are joined, shortened and rewritten.

Methods and Materials section are divided into subsections, including e.g.: Study Design, Participants, Questionnaire, Statistical Analysis.

Methods are described. 

All abbreviations are explained when they appear for the first time in the text.

We provided research results based on the data obtained from the questionnaire.

In conclusions, we put amendments regarding the purpose as well as the methodology, discussions, results and conclusions.

Please, see document with track changes. 

Reviewer 2 Report

This is an interesting paper and worthwhile to understand the barriers to successful weight loss in Lebanese culture. The statistical analysis is good and detailed however other aspects of the paper are significantly less detailed. More clarity regarding the aims, design and hypotheses of the research need to be stated more clearly from the outset. 

Abstract

Line 31: avoid term “better body shape”

Line 33: check English “more perceived among women”. More prevalent instead?

Introduction

Line 45: “The most common reason for diet failure is attributed to stress and food deprivation”. Please provide additional references to assert this claim.

Line 55: Can you add references for these intervention.

Line 58: Paragraph jumps to effect of stress. Consider rewriting first line of this paragraph.

Line 68: The final paragraph needs to be clearer on what this study is. I.e. This is a literature review of weight loss interventions… At this point, as the reader I am not sure what the paper is reviewing.

Literature review

This whole section feels as though it should be part of the introduction. There should be a description of how the literature review was carried out.

The structure needs some work e.g. there are one sentence paragraphs that appear to make one isolated point before moving onto a new point (i.e. Line 112).

Materials and Methods

The statistical methods of questionnaire development are good, but again, I wasn’t expecting this section as the introduction suggests the paper is a review.

Line 137: How were the interviewers picked at random?

Who are the participants?

Results

The hypotheses should be defined earlier in the paper. 

Conclusion

The first line of the conclusion is very clear what the research question is which is good; this clarity is needed in earlier sections of the paper.

Author Response

Dear Reviewer, 

thank you for your comments. We recognise the value that is added. 

We avoid term “better body shape”. 

We changed “more perceived among women” to more prevalent instead. 

“The most common reason for diet failure is attributed to stress and food deprivation”. We provided additional references to assert this claim.

Line 55: We add references for these intervention.

Line 58: We were rewriting first line of this paragraph.

Line 68: The final paragraph we made clearer on what this study is. I.e. This is a literature review of weight loss interventions… 

We added description of how the literature review was carried out.

How were the interviewers picked at random?

We explained:

Who are the participants?

The hypotheses are defined earlier in the paper.

Please, find attached the corrected version with track changes. 

Round 2

Reviewer 1 Report

Review of the revised version of the Manuscript applsci-922188

Title: Comprehensive Approach in Medical Nutrition Therapy for Adults’ Weight Loss Management in Lebanon

Dear Authors,

I appreciate your prompt response to the comments in the review and the improvement of the article. However, despite the declaration of improvement, the article needs additional redrafting, including linguistic improvement. Furthermore, the Authors state that the work is presenting  the comprehensive approach in Medical Nutrition Therapy for adults’ weight loss management, but in my opinion it was not presented in this revised work.

I have the following comments to the presented article:

The lack of line numbers makes difficult to accurately indicate the fragment to be corrected.

Featured Application: I still maintain that it does not add anything to the article.

Introduction:

  • Please check again this part of the paper.
  • Some sentences are repeated:

(Introduction line 4) Alternatively, elements and variables that actively interfere with MNT are to be determined based on their effect on obesity financially and economically.

(Introduction line 4 from the bottom) Alternatively, elements and variables that actively interfere with MNT shall be assessed and determined based on their effect on obesity.

(Introduction line 10) However, the proportions vary from country to country and within geographical areas in the same country [1, 2].

(Introduction line 23 from the bottom) However, the proportions vary from country to country and within geographical areas in the same country [1, 2].

Materials and Methods

  • the model of data analysis procedures should be numbered figure 1;
  • in table 1 please explain what does mountain as home address /work address mean?

Results

  • this section contains the results described in the text. This makes their interpretation difficult. I suggest to present the results in the form of a table, and to provide a short description in the text;
  • it is still unclear what was the purpose of formulating the seven research hypothesis and how does it relate to the development of the MNT?

 Discussion

  • the last part of the Discussion section is rather methodology.

In conclusion, this article requires major amendments including English proofreading.  

Author Response

Dear reviewer,

We are grateful for your suggestions for improvement.

1. We checked that sentences are not repeated:

(Introduction line 4) Alternatively, elements and variables that actively interfere with MNT are to be determined based on their effect on obesity financially and economically.

(Introduction line 4 from the bottom) Alternatively, elements and variables that actively interfere with MNT shall be assessed and determined based on their effect on obesity.

(Introduction line 10) However, the proportions vary from country to country and within geographical areas in the same country [1, 2].

(Introduction line 23 from the bottom) However, the proportions vary from country to country and within geographical areas in the same country [1, 2].

2. The model of data analysis procedures is numbered in figure 1.

3. Please explain what does mountain as home address /work address mean? The address of the participants is divided between city, village, and mountain, differing on the geographical scale. The city is on the coast, and the mountain is where it snows in winter, and the village is in between.

4. It is still unclear what was the purpose of formulating the seven research hypothesis, and how does it relate to the development of the MNT? The seven hypotheses that were analysed regarding the research subject aim at answering the main question regarding the variables directing Medical Nutrition Therapy (MNT) for adult weight loss in Lebanon, which are presented in the final illustration of each of the seven accepted hypotheses via AMOS software

5. Discussion - the last part of the Discussion section is rather a methodology. We have merged the chapters Results and Discussion and add additional text for discussion.

6. The article is proofread by a native speaker.

Reviewer 2 Report

Thank you for your response. I feel the paper has improved with these changes and recommend accepting. Please proof read to avoid grammatical (e.g. line 95)and spelling errors before final submission. 

Author Response

Thank you for your review. We corrected spelling mistakes.

Round 3

Reviewer 1 Report

Present hypotheses in Table. 

Author Response

Thank you for your review. Please, find attached the corrected version of the manuscript.